# Exploring the prevalence of Human Papillomavirus (HPV) genotypes in PAP smear samples of women in northern region of United Arab Emirates (UAE): HPV Direct Flow CHIP system-based pilot study

Heba Issa Odeh[1], Sara Rashid Al-badi[1], Basma Karima[1], Takrim Abdulwali Saeed[1], Nazeerullah Rahamathullah[1,2‡]*, Eman Hassan Ibrahim[1]*, May Khalil Ismail[1]*, Zahra Arshad Khan[1,2]

1 Department of Biomedical Sciences, College of Medicine, Gulf Medical University, Ajman, United Arab Emirates, 2 Thumbay Research Institute for Precision Medicine, College of Medicine, Gulf Medical University, Ajman, United Arab Emirates

☯ These authors contributed equally to this work.
‡ This author also contributed equally to this work.
* dr.nazeer@gmu.ac.ae (NR); dr.eman@gmu.ac.ae (EHI); dr.maykhalil@gmu.ac.ae (MKI)

## Abstract

### Objective

The aim of this study was to explore the prevalence of low and high-risk HPV genotypes in PAP smear samples of women in northern region of the UAE using HPV direct flow CHIP method.

### Methods

A cross-sectional retrospective study was conducted between September 2021 to April 2022. A total of 104 liquid-based cervical cytology samples were obtained from women aged 20–59 years attending the Gynaecology out-patient department of Thumbay University Hospital and other hospitals of Northern Emirates of UAE, processed for the routine cytological examination to identify and differentiate morphological changes of the PAP smear samples. HPV genotyping was performed using HPV direct flow CHIP method.

### Results

In total, 112 HPV genotypes were detected in 63 women (60.57%) included 18 abnormal cytological and 45 normal epithelial samples. 63 LR and 49 HR HPV genotypes were identified in all the 63 positive samples. Highest rate of infection with multiple LR and HR HPV genotypes were detected in women aged 40–49 years (25.9%) and 20–29 years (23.5%). Infection by HPV6 (13.46%), HPV11 (9.61%), HPV16 (9.61%), HPV62/81 (7.69%) and HPV45 (7.69%) were the most common genotypes. A moderate increase than expected incidence of HPV45 and 62/81 (7.69%) were detected. Co-infection with multiple low and

**Data Availability Statement:** All relevant data are within the manuscript and its Supporting Information files.

**Funding:** The author(s) received no specific funding for this work.

**Competing interests:** The authors have declared that no competing interests exist.

high-risk genotypes is present in 20.2% cases; in that, HPV6 (15.9%) was the most common followed by HPV62/81 (12.7%) and HPV16 (11.11%). The prevalence of HPV18 was found to be 1.6%.

## Conclusion

The genotypes 6, 45, 16, 11, 67, 62/81 were the most common HPV infections in the women between the age group of 21 and 59-years-old. A moderate increase of HPV45, 62/81 and much less prevalence of HPV18 were detected in the study population. 43.27% of the normal epithelia were positive to different low and high-risk HPV genotypes. This finding highlights the importance of molecular genotyping of HPV to emphasize the cervical screening triage.

## Introduction

Cervical cancer is the second most common cancer among women worldwide [1]. According to the World Health Organization (WHO), there were an estimated 604,000 new cases of cervical cancer and 342,000 deaths from the disease worldwide in 2020. This makes cervical cancer the fourth most common cause of cancer death among women globally [2, 3]. It ranks as the 5th most frequent cancer among the women in the United Arab Emirates and the 3rd most frequent cancer among women between 15–44 years of age. It is the 13th leading cause of cancer-related mortality in the UAE [4]. Human Papillomavirus (HPV) is transmitted through sexual mode and has been identified as a causative agent of cervical cancer [5, 6]. To date, more than 450 genotypes of human papillomaviruses (HPVs) have been identified [7], 40 of which infect the genital tract and many of them have been found in cervical cancers, while others are found rarely or not at all in large series of cancers, which gives rise to the nomenclature of 'high-' and 'low-risk' HPVs [8, 9]. A persistent infection with a high-risk HPV contributes to the development of invasive cervical cancer [10].

HPV infection is reported in about 80% of the cases with low-grade squamous intraepithelial lesions (LSILs) and 90% of cases with high-grade squamous intraepithelial lesions (HSILs) [11]. The crude incidence rate of HPV related cancer in UAE is 4.03/100,000 population. There is no clear data available on the low and High-risk HPV burden in the general population of UAE. However, in Western Asia, the region United Arab Emirates belongs to, about 2.5% of women in the general population are estimated to harbor cervical HPV-16/18 infection at a given time, and 72.4% of invasive cervical cancers are attributed to HPVs 16 or 18 [4]. Some studies reported incidence of cervical abnormalities in cervical smears among women with lesions including LSIL, HSIL, ASCUS (atypical squamous cells of undetermined significance) and glandular abnormalities [12]. Though the reports revealed confirmation of HPV infection and graded the biopsy tissue or pap smeared cells, the low and high-grade HPV genotypes were not clearly concluded except some high-risk HPV genotypes. High-risk HPV infection, the single most important etiologic agent for uterine cervical carcinomas, appears to show regional variation in the prevalence of its subtypes [13].

Worldwide the most common HPV genotypes are HPV 16/18 detected in up to 70% of cervical cancers [14]. 17.8% HPV infection rate was reported among woman in Oman, and it is much higher than in studies from neighboring countries such as Iran (7.8%), Qatar (6.1%) and KSA (9.8%) that was conducted on normal and abnormal cytology [15]. Some studies reported

that the high-risk HPV genotypes are available from the countries with somewhat similar geography and demography to the UAE. Limited number of studies are available on some high-risk HPV16, 31 & 18, but no detailed studies on genotyping of other 32 high and low risk genotypes (HR HPV26, 33, 35, 39, 45, 51, 52, 53, 56, 58, 59, 66, 68, 73 & 82, and LR HPV6, 11, 40, 42, 43, 44, 54, 55, 61, 62, 67, 69, 70, 71, 72, 81 & 84) have yet been performed for in the UAE population [13]. So, the present study aimed to explore the prevalence of HPV infection and its genotypes in women with normal and abnormal uterine cervices in the Northern region of UAE. Determining the prevalence of HPV infection and its genotypes in PAP smear samples is critical for effective prevention, early detection, and management of HPV-related diseases, as well as for public health awareness and initiatives.

## Methods

### Ethical approval

The present study was approved by the Institutional Review Board (IRB) of Gulf Medical University, Ajman, UAE. The approval letter from the IRB, dated October 14, 2021, indicated that the study had already received approval in the IRB meeting of September 2021 (Ref.no.IRB/ COM/STD/98/Oct.2021). The study design and protocol followed the Good Clinical Practice (GCP) guidelines 2021, National Drug Abuse Treatment Clinical Trials Network.

### Study design

A cross-sectional retrospective study was conducted on the liquid-based cervical cytology samples received from various hospitals of Northern Emirates of the UAE to the Department of Pathology at Thumbay Laboratory, Thumbay University Hospital (TUH), Ajman, UAE.

### Data collection

In accordance with the IRB approval, the data of the study samples were retrospectively obtained from the medical records of Thumbay Laboratory, TUH, starting the period from September 01, 2021, to April 30, 2022.

### Study populations

All the Liquid-based cervical cytology samples were obtained from women aged 20–59 years attending Gynaecology out-patient department of TUH and other hospitals of Northern Emirates of the UAE. All the samples were collected by gynecologists using Sure Path® liquid-based cytology system (Tri Path Imaging, Burlington, NC) and sent to the department of pathology at Thumbay Laboratory, TUH to diagnose the cervical abnormalities and detect the positivity of HPV and their genotypes. Upon arrival, the samples were processed for cytological examination as per the laboratory protocol [16].

### Cytological examination

Papanicolaou stained thin prep liquid-based PAP smeared slides (Thinprep 2000 tissue processor) were reviewed by qualified pathologists to confirm the diagnosis of cytological abnormalities. The diagnosed slides were classified based on morphologic criteria for cervical neoplasia according to the Bethesda identification system 2014 [16]. All the PAP smear slide results and demographic data (Nationality and age) of the patients were retrieved from the medical records of Thumbay laboratory, TUH.

### HPV detection and genotyping

The residual concentrated materials in the Thinprep concentrated tubes were used for HPV DNA extraction followed by detection of high & low-risk HPV genotypes by DNA hybridization by HPV Direct Flow CHIP method (Vitro Mast Diagnostica, Spain). The HPV Direct Flow CHIP is optimized for the direct use of clinical samples for DNA extraction, polymerization and flow through reverse hybridization.

### Sample preparation for PCR

30μl of homogenized liquid cytology sample suspension was directly used for polymerization of HPV genes without any separate DNA extraction procedure. The PCR master mix contains buffer, dNTPs (U/T), DNase/RNase-free water, biotinylated primers, DNA polymerase, and UNG. Primers included a specific amplification of a fragment of the region L1 of the HPV and 35 HPV genotype primers for high risk (16, 18, 26, 31, 33, 35, 39, 45, 51, 52, 53, 56, 58, 59, 66, 68, 73 and 82) and low risk (6, 11, 40, 42, 43, 44, 54, 55, 61, 62, 67, 69, 70, 71, 72, 81 and 84) strains and a specific primer for the amplification of a human genomic DNA fragment (beta-globin gene) an internal control to show adequacy of the sample. A specific PCR program cycle was followed by the protocol recommended by the manufacturer (Vitro Mast Diagnostica, Spain).

### Flow-through reverse hybridization

Following PCR amplification, biotinylated amplicons were hybridized on membranes containing multiple probes, each for a specific target genotype, along with control probes for amplification and hybridization. The DNA-Flow technology allowed faster binding of PCR products to their specific probes compared to conventional surfaces, thus speeding up hybridization. The HPV Direct Flow Chip kit offers two lyophilized formats for analyzing patient samples, each containing reagents for multiplex PCR amplification and hybridization of 24/48 clinical samples. Upon hybridization of specific amplicons with their corresponding probes, the signal was detected via an immunoenzymatic colorimetric reaction with Streptavidin-Phosphatase and a chromogen (NBT-BCIP), forming insoluble precipitates at the positions of hybridization on the membrane. The software HybriSoftTM automatically analyzed the results, with all probes duplicated to ensure the validity of the automatic analysis. The system was validated using fresh DNA purified from clinical samples with the MagNa Pure (Roche) [17].

### Statistical analysis

Data were analyzed using the statistical package SPSS software version 28.0. Descriptive data were represented as mean ± SD. Pearson's chi-square test was performed in Tables 2, 3 & 6 with $p$-value <0.05 considered as statistically significant.

## Results

### Cytological examination

A total of 104 cervical smears samples were collected and processed for the routine cytopathological examination to identify and differentiate the morphological changes of the PAP smear samples. Based on Bethesda system 2014, the samples were graded as ASCUS identified in 13 (12.5%) samples, LSIL accounted for 7 (6.7%) samples, 1 case (0.96%) was HSIL and a single sample (0.96%) was reported as atypical squamous cells cannot rule out high grade squamous intraepithelial lesion (ASC-H) and normal epithelia were identified in 82 (78.84%) PAP smear samples and their details are presented in Table 1; Figs 1 and 2.

Table 1. Grading of PAP smear samples collected from women with different age groups.

| Grading of Cervical cytology | Different age groups | | | | Total n = 104(%) |
|---|---|---|---|---|---|
| | 20–29 *n = 34 (%)* | 30–39 *n = 44 (%)* | 40–49 *n = 23(%)* | 50–59 *n = 04 (%)* | |
| ASCUS | 7 (53.8%) | 5 (38.5%) | 1 (7.7%) | 0 | 13 (12.5%) |
| LSIL | 1 (14.3%) | 4 (57.14%) | 2 (28.6%) | 0 | 7 (6.7%) |
| HSIL | 0 | 0 | 1 (100%) | 0 | 1 (0.96%) |
| ASC-H | 0 | 1 (100%) | 0 | 0 | 1 (0.96%) |
| Normal epithelia | 26 (31.7%) | 33 (40.24%) | 19 (23.2%) | 04 (4.9%) | 82 (78.84%) |

*n*–number of samples

## HPV detection and genotyping

Out of 104 liquid biopsy cervical samples, 63 were positive to HPV infection and 41 samples were negative to HPV infection and it is mentioned in Table 2. In total, 112 HPV genotype were detected in the 63 pap smear samples, among them, 32 genotypes were identified in 18 abnormal cytology samples (ASCUS = 10, LSIL = 6, HSIL = 1 & ASC-H = 1) and 80 genotypes were detected in 45 normal epithelial samples. In the abnormal cytology samples, 18 HPV genotypes were identified from the positive ASCUS samples. Among them, HR-HPV 16 were

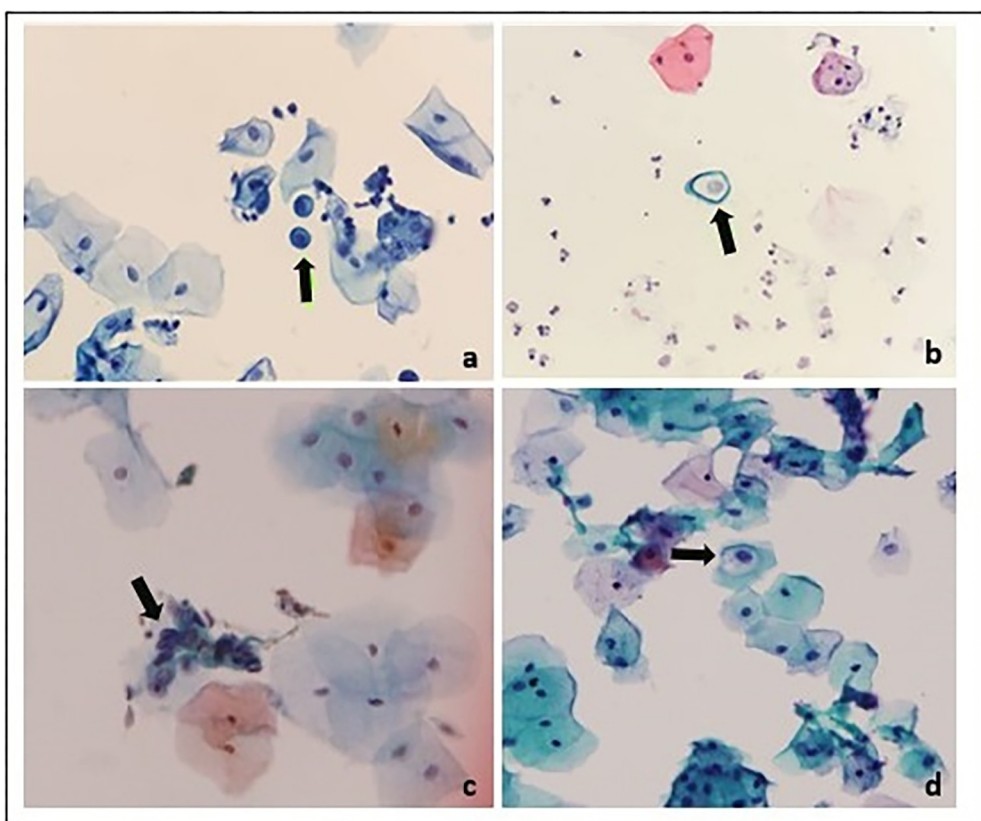

**Fig 1. Grading of PAP smeared cytological samples (400X magnification, olympus BX51). a.** Atypical squamous cells, cannot rule out high grade squamous intraepithelial cells (ASC-H) **b.** Atypical squamous cells of undetermined significance (ASCUS), **c.** High grade squamous intraepithelial lesion (HSIL), **d.** Low grade squamous intraepithelial lesion (LSIL).

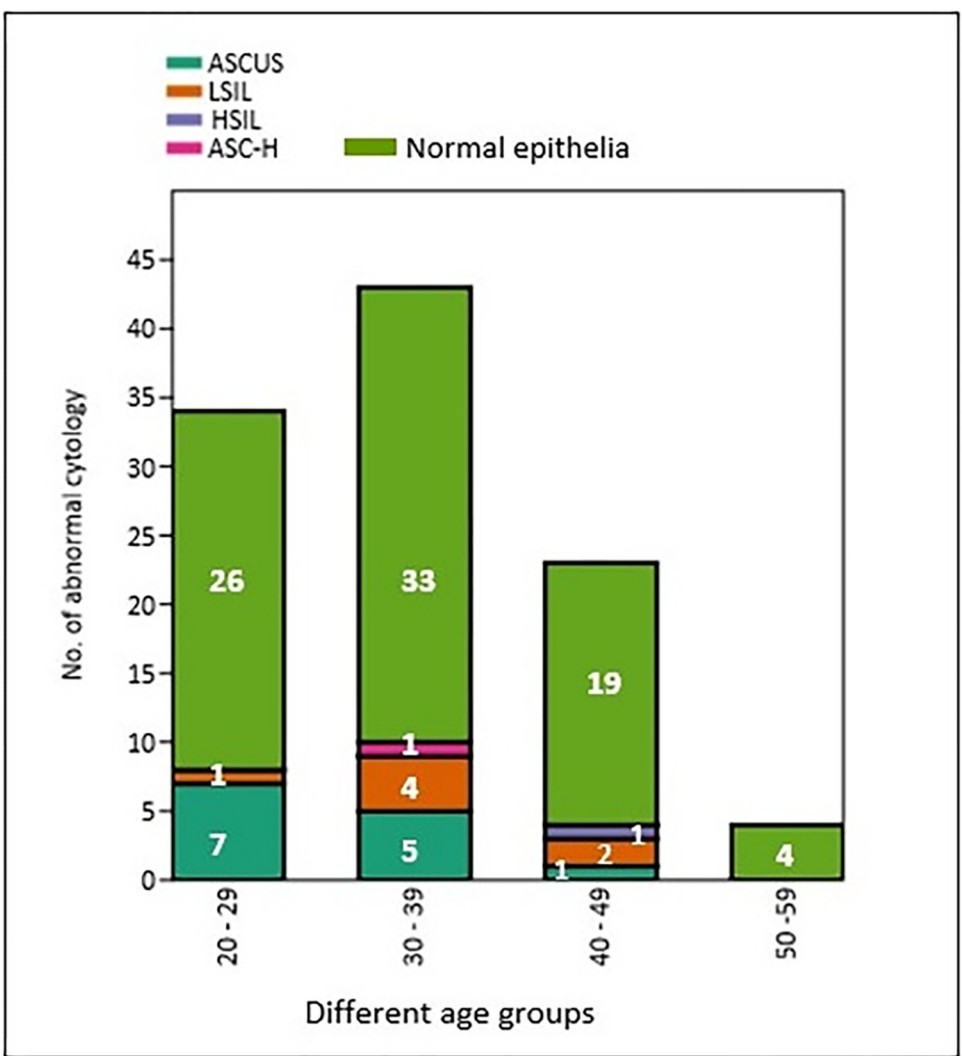

**Fig 2. Cytology grading of PAP smear samples collected from different age groups of women.**

**Table 2. Detection of HPV in different abnormal cytological and normal epithelial samples.**

| Cytological grading n = 104 (%) | Detection of HPV n = 104 (%) | | Pearson Chi-Square p value |
|---|---|---|---|
| | Positive n = 63 (60.57%) | Negative n = 41(39.43%) | |
| ASCUS | 10 (9.61%) | 3 (2.89%) | 0.019 |
| n = 13 (12.5%) | | | |
| LSIL | 6 (5.77%) | 1 (0.93%) | |
| n = 7 (6.7%) | | | |
| HSIL | 1 (0.96%) | 0 | |
| n = 1 (0.96%) | | | |
| ASC-H | 1 (0.96%) | 0 | |
| n = 1 (0.96%) | | | |
| Normal epithelia | 45 (43.27%) | 37 (35.57%) | |
| n = 82 (78.84%) | | | |

n—number of samples

detected in 3 samples and the LR genotypes 11 & 70 were detected in 2 samples. The HR-HPV 18 was detected in a sample along with other 4 genotypes including 16. In the 6 LSIL samples, in total, 10 HPV genotypes were detected. Among them, the LR-HPV 6 & 16 were detected in 2 samples and in the case of HSIL, and ASC-H cytology, the LR-HPV 62/81, 35 & HR-HPV 67,16 were detected. In normal cytology, the genotype HPV6, 11, 45, 68, 70, 82 were detected in 11, 8, 7, 3 and 2 samples respectively and the HPV31 and 16 were encountered in 4 different samples. Number of single and multiple, low and high-risk genotypes detected in different cytology samples are given in Table 3 and in Fig 3.

**Table 3. Number of single, multiple low and high-risk HPV genotypes in different cytology samples.**

| HPV positive cervical cytology samples n = 63 | | Identified HPV genotypes | | | | | Pearson Chi-Square p value |
|---|---|---|---|---|---|---|---|
| | | Single LR-HPV n = 12 (%) | Single HR-HPV n = 13 (%) | Multiple LR -HPVs n = 12 (%) | Multiple HR-HPVs n = 5 (%) | Multiple LR & HR-HPVs n = 21 (%) | |
| Abnormal cervical cytology samples with HPV infection n = 18 | ASCUS n = 10 | HPV 11 HPV40 n = 2 (1.92%) | HPV58 HPV31 HPV16 n = 3 (2.88%) | HPV6,70 n = 1 (0.96%) | HPV52,86 n = 1 (0.96%) | HPV67,16 HPV11,45 HPV54,70,18,16,33 n = 3 (2.88%) | 0.147 |
| | LSIL n = 6 | 0 | HPV16 HPV66 HPV51 n = 3 (2.88%) | HPV6,43 n = 1 (0.96%) | HPV45,59, 82 n = 1 (0.96%) | HPV6&16 n = 1 (0.96%) | |
| | HSIL n = 1 | 0 | 0 | 0 | 0 | HPV62/81,35 n = 1 (0.96%) | |
| | ASC-H n = 1 | 0 | 0 | 0 | 0 | HPV67,16 n = 1 (0.96%) | |
| Normal epithelia n = 45 | | HPV6 (n = 4) HPV11 (n = 4) HPV84 HPV70 n = 10 (9.61%) | HPV59 HPV73 HPV18 HPV45 HPV68 HPV82 HPV58 n = 7 (6.73%) | HPV6/43 HPV62/81 (n = 3) HPV43 & 62/81 HPV6,11&42 HPV6&11 HPV6 & 67 HPV6&70 HPV42, 44/55 n = 10 (9.61%) | HPV56& 68 HPV31/68 HPV16& 45 n = 3 (2.88%) | HPV11,67,16 HPV6,42,51 HPV62/81,84,39,58 HPV11,82 HPV61,45 HPV61,31,51 HPV54,67,16,45 HPV61,39,45 HPV70,31 HPV67,16 HPV6,39 HPV62/81,67,31 HPV44/55,54,45,68 HPV54,58 HPV62/81,33,45,68 n = 15 (14.42%) | |

*n*–number of samples, LR–Low risk, HR–High-risk

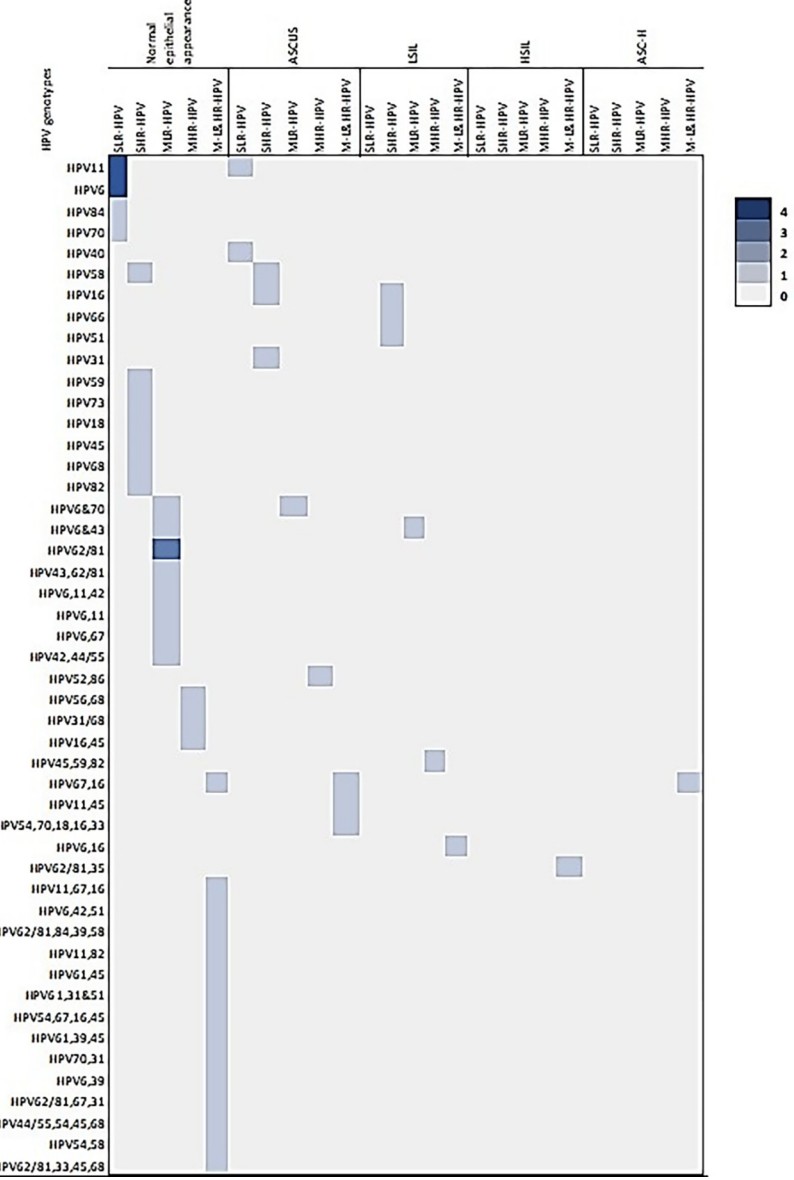

**Fig 3. Frequency of single low, high-risk genotypes, multiple low, high-risk and multiple both low and high-risk genotypes in the normal epithelial cells, ASCUS, HSIL, LSIL, and ASC-H of PAP smear samples.**

## Age group and ethnicity of the HPV positive patient

A total of 4 different age groups were concluded in this study population. Among them, 34 (32.7%) patients aged between 20–29 years, 39 (37.5%) patients fall in the 30–39 age group, the 40–49 age group included 27 (26%) patients, and 4 (3.8%) patients were the age group of 50–59-year-old and these details are given in the Tables 1 and 4. All the patients were grouped in to two ethnicities: Arab and non-Arab. A total of 54 patients were Arab which accounted for 51.9%; among them 31 (49.2%) tested positive to HPV and 23 (56.1%) were negative. Fifty cases (48.1%) were grouped under non-Arab; in that, 32 (50.8%) tested positive and 18 (43.9%) were negative to HPV infection. Detailed descriptive data of the patients' age group, ethnicity, HPV positivity, single HPV genotype detection with low risk, high risk genotypes and multiple genotypes with low risk and high-risk genotypes are presented in the Table 4.

**Table 4. Frequency of single, multiple low and high-risk HPV genotypes with different age group of both Arab and non-Arab study population.**

| Patient's Age group n (%) | Ethnicity n | HPV result +ve/-ve n | HPV genotypes | | | | |
|---|---|---|---|---|---|---|---|
| | | | Single genotype | | Multiple genotypes | | |
| | | | LR-HPV n (%) | HR-HPV n (%) | Multiple LR-HPVs n (%) | Multiple HR-HPVs n (%) | Multiple LR & HR HPVs n (%) |
| **20–29** n = 34 (32.7%) | Ar n = 17 | +ve n = 6 | HPV11 n = 1 (1.6%) | 0 | HPV42, 44/45 n = 1 (1.6%) | HPV31/68 n = 1 (1.6%) | HPV11/67,16 HPV11,45 HPV54,58 n = 3 (4.8%) |
| | | -ve n = 11 | - | - | - | - | - |
| | NAr n = 17 | +ve n = 12 | HPV11 HPV84 n = 2 (3.2%) | HPV58 HPV16 HPV73 HPV58 n = 4 (6.3%) | 0 | HPV16,45 n = 1 (1.6%) | HPV67,16 HPV61,39, 45 HPV62/81,67,31 HPV54,70, 18,16,33 HPV62/81,33,45,68 n = 5 (7.9%) |
| | | -ve n = 5 | - | - | - | - | - |
| **30–39** n = 39 (37.5%) | Ar n = 18 | +ve n = 12 | HPV11 HPV6 n = 2 (3.2%) | HPV16 HPV51 n = 2 (3.2%) | HPV6,43 HPV6,11 HPV6,67 HPV6,70 n = 4 (6.3%) | HPV52/86 HPV45,59,82 n = 2 (3.2%) | HPV61,45 HPV54,67, 16,45 n = 2 (3.2%) |
| | | -ve n = 6 | - | - | - | - | - |
| | NAr n = 21 | +ve n = 13 | HPV6 HPV6 HPV40 n = 3 (4.8%) | HPV59 HPV66 n = 2 (3.2%) | HPV6,70 HPV6,43 HPV62/81 n = 3 (4.8%) | HPV56,68 n = 1 (1.6%) | HPV70,31 HPV44/55,54,45,68 HPV67,16 HPV62/81,84,39,58 n = 4 (6.3%) |
| | | -ve n = 8 | - | - | - | - | - |
| **40–49** n = 27 (26%) | Ar n = 17 | +ve n = 13 | HPV6 HPV70 HPV11 n = 3 (4.8%) | HPV18 HPV45 HPV68 n = 3 (4.8%) | HPV62/81 HPV43, 62/81 n = 2 (3.2%) | 0 | HPV6,42,51 HPV6,16 HPV11,82 HPV61,31, 51 HPV67,16 n = 5 (7.9%) |
| | | -ve n = 4 | - | - | - | - | - |
| | NAr n = 10 | +ve n = 6 | 0 | HPV82 HPV31 n = 2 (3.2%) | HPV62/81 HPV6,11, 42 n = 2 (3.2%) | 0 | HPV6,39 HPV62/81,35 n = 2 (3.2%) |
| | | -ve n = 4 | - | - | - | - | - |

(*Continued*)

**Table 4.** (Continued)

| Patient's Age group n (%) | Ethnicity n | HPV result +ve/-ve n | HPV genotypes | | | | |
|---|---|---|---|---|---|---|---|
| | | | Single genotype | | Multiple genotypes | | |
| | | | LR-HPV n (%) | HR-HPV n (%) | Multiple LR-HPVs n (%) | Multiple HR-HPVs n (%) | Multiple LR & HR HPVs n (%) |
| 50–59 n = 4 (3.8%) | Ar n = 2 | +ve n = 0 | - | - | - | - | - |
| | | -ve n = 2 | - | - | - | - | - |
| | NAr n = 2 | +ve n = 1 | HPV11 n = 1 (1.6%) | 0 | 0 | 0 | 0 |
| | | -ve n = 1 | - | - | - | - | - |

n–number of samples, Ar–Arab, NAr–Non-Arab, +ve/-ve–Positive/Negative, LR–Low-risk, HR–High-risk

In the age group of 20–29-year-old Arab ethnicity, 10 HPV genotypes were detected, among them HPV11 & 45 were detected in 3 (30%), 2 (20%) samples and in age group of 30–39-year-old, 22 genotypes were observed, the HPV6 (23.8%) was common genotype followed by HPV 45 (14.3%). There are 21 HPV genotypes that were detected in the age group of 40–49-year-old of the same ethnicity. Among them, HPV6 was a common genotype (14.3%) and then HPV16, 11 and 62/81 (9.52%) were detected and these details are given in Figs 3–5.

In the non-Arab ethnicity, 25 genotypes were detected in the age group of 20–29-years old. Among them, the HR-HPV16 (16%) was predominant and the HPV45 (12%), 62/81, 67, 33 (8%) were next to it and the HR-HPV18 & 16 were identified together in one sample along with HPV54, 70 and 33. In the age group of 30–39-year-old, 24 genotypes were detected, the HPV6 (16.6%) was frequently identified and then the HPV70, 68, 62/81 were detected only in 8.3% samples. The HPV16 was detected in one sample with HPV67. 10 HPV genotypes were detected in the age group of 40–49-year-old of the same ethnicity, among them 20% were HPV6 & 62/81 and the genotype HPV11 was detected in one sample of the age group 50–59-year-old of the same ethnicity and these details are mentioned in the Figs 3–5 and Table 4.

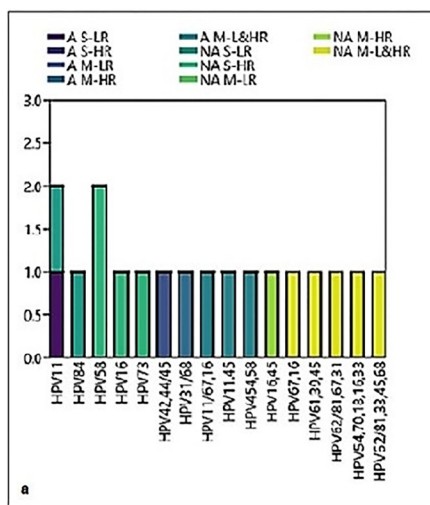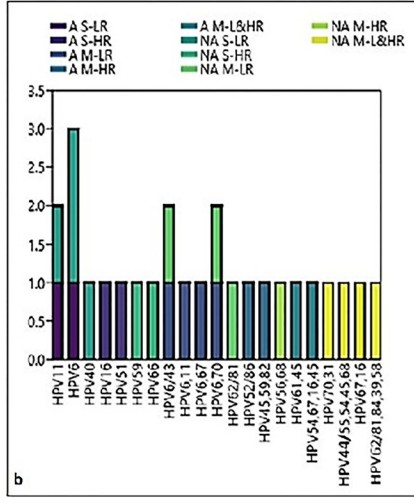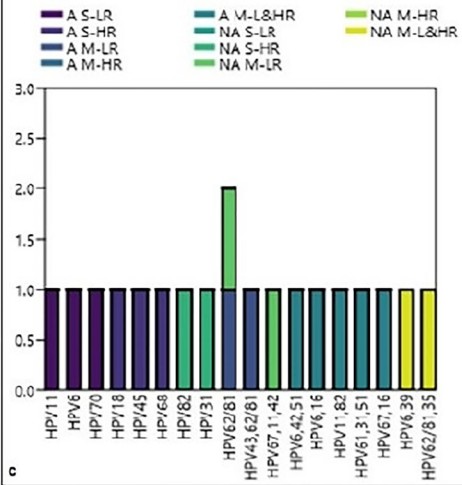

**Fig 4.** Frequency of single, multiple low and high-risk HPV genotypes among the different age groups of Arab and non-Arab study population a) Age group 20–29; b) 30–39; c) 40–49.

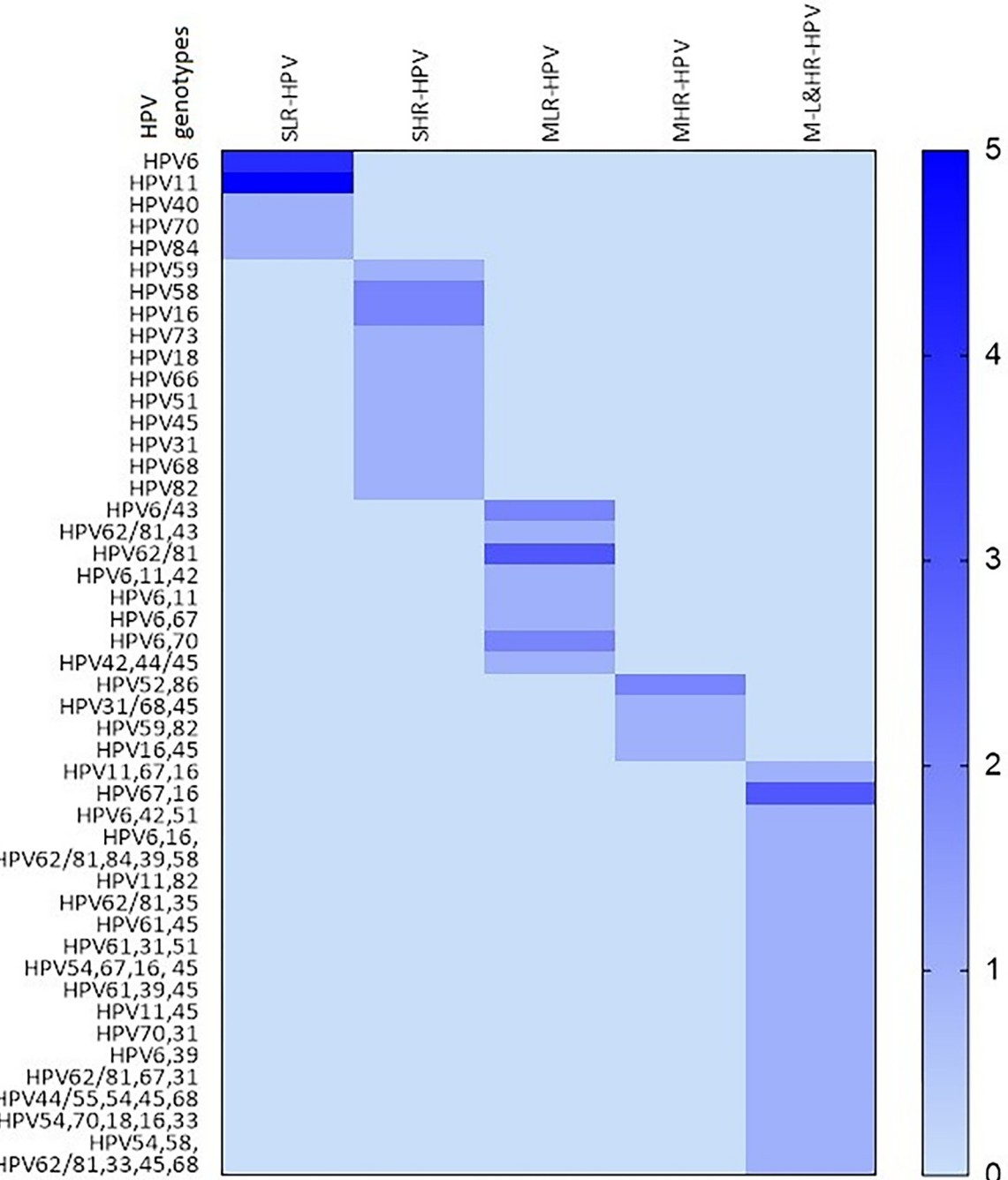

**Fig 5. Frequency of identified single, multiple low & High-risk HPV genotypes among the study population.**

Among the 63 HPV positive samples, 25 (39.68%) samples were infected by single HPV genotypes, of which 12 (19.04%) were infected by only low-risk genotypes and whereas high-risk single genotypes detected in 13 (20.63%) samples. Multiple genotypes were detected in 38 (60.31%) samples; in that, 12 (19.04%) were infected with low-risk genotypes, 5 (7.93%) were with high-risk genotypes and 21 (33.33%) samples were infected by multiple genotypes of both low and high-risk types and these details are given in the Table 5 and Fig 5. Number of single,

**Table 5. Frequency of single, multiple low and high-risk HPV genotypes detected in all the HPV positive PAP smear samples.**

| Genotype of HPV *n (%)* | Low risk /High-risk genotype | HPV genotypes | No. of samples *(n = 104)* | Total *(%)* |
|---|---|---|---|---|
| **Single genotype *n = 25 (24.04%)*** | LR genotype | HPV6 | 4 | 12 *(11.54%)* |
| | | HPV11 | 5 | |
| | | HPV40 | 1 | |
| | | HPV70 | 1 | |
| | | HPV84 | 1 | |
| | HR genotype | HPV59 | 1 | 13 *(12.5%)* |
| | | HPV 58 | 2 | |
| | | HPV16 | 2 | |
| | | HPV73 | 1 | |
| | | HPV18 | 1 | |
| | | HPV66 | 1 | |
| | | HPV51 | 1 | |
| | | HPV45 | 1 | |
| | | HPV31 | 1 | |
| | | HPV68 | 1 | |
| | | HPV 82 | 1 | |
| **Multiple genotypes*n = 38 (36.54%)*** | Multiple Low risk genotypes | HPV6/43 | 2 | 12 *(11.54%)* |
| | | HPV62/81,43 | 1 | |
| | | HPV62/81 | 3 | |
| | | HPV6,11,42 | 1 | |
| | | HPV6,11 | 1 | |
| | | HPV6,67 | 1 | |
| | | HPV6,70 | 2 | |
| | | HPV42,44/55 | 1 | |
| | Multiple high-risk genotypes | HPV52,86 | 2 | 5 *(4.8%)* |
| | | HPV31/68,45 | 1 | |
| | | HPV45,59,82 | 1 | |
| | | HPV16,45 | 1 | |
| | Multiple low & high-risk genotypes | HPV11,67,16 | 1 | 21 *(20.2%)* |
| | | HPV67,16 | 3 | |
| | | HPV6,42,51 | 1 | |
| | | HPV6,16, | 1 | |
| | | HPV62/81,84,39,58 | 1 | |
| | | HPV11,82 | 1 | |
| | | HPV62/81,35 | 1 | |
| | | HPV61,45 | 1 | |
| | | HPV61,31,51 | 1 | |
| | | HPV54, 67,16, 45 | 1 | |
| | | HPV61,39,45 | 1 | |
| | | HPV11,45 | 1 | |
| | | HPV70,31 | 1 | |
| | | HPV62/81,67,31 | 1 | |
| | | HPV44/55,54, 45,68 | 1 | |
| | | HPV54,70,18,16,33 | 1 | |
| | | HPV54,58, | 1 | |
| | | HPV62/81,33,45,68 | 1 | |

*(Continued)*

**Table 5.** (Continued)

| Genotype of HPV *n (%)* | Low risk /High-risk genotype | HPV genotypes | No. of samples *(n = 104)* | Total *(%)* |
|---|---|---|---|---|
| **Negative *n = 41 (39.42%)*** | - | - | - | - |

*n*–number of samples

**Table 6. Frequency of single, multiple low and high-risk HPV genotypes with different age groups of the study population.**

| Age Groups *n = 104* | HPV genotypes | | | | | | Pearson's Chi-square *p* value |
|---|---|---|---|---|---|---|---|
| | Single strain | | Multiple strains | | | -ve *n = 41* (%) | |
| | Low risk HPVs *n = 12 (%)* | High risk HPVs *n = 13 (%)* | Low risk HPVs *n = 12 (%)* | High risk HPVs *n = 5 (%)* | Multiple LR& HR HPVs *n = 21 (%)* | | |
| **20–29 *n = 34*** | 3 (8.8%) | 4 (11.8%) | 1 (2.9%) | 2 (5.9%) | 8 (23.5%) | 16 (47.1%) | 0.577 |
| **30–39 *n = 39*** | 5 (12.8%) | 4 (10.3%) | 7 (17.9%) | 3 (7.7%) | 6 (15.4%) | 14 (35.9%) | |
| **40–49 *n = 27*** | 3 (11.1%) | 5 (18.5%) | 4 (14.8%) | 0 | 7 (25.9%) | 8 (29.6%) | |
| **50–59 *n = 4*** | 1 (25%) | 0 | 0 | 0 | 0 | 3 (75%) | |

*n*–number of samples, -ve–Negative, LR–Low-risk, HR–High-risk

multiple low and high-risk HPV genotypes with different age group of women are given in Table 6. Prevalence of the low and high-risk genotypes HPV6 (13.46%), HPV11 (9.61%), HPV62/81 (7.69%), HPV16 (9.61%) and HPV45 (7.69%) were moderately high in this finding.

## Discussion

Cervical cancer is the most common malignancy among middle-aged **women** [18] and recently the HPV testing along with PAP screening is an important diagnostic and prognostic procedure to find out various genotypes that belongs to either high risk or low risk group. Therefore, screening of HPV in PAP smear sample is a vital procedure to know the genotypes in various diverse regions.

The present study collected 104 PAP smear samples, of which 21.16% samples showed abnormal cytology that is precancerous type and other samples (78.84%) appeared normal cytology. In recent years, the rate of identified abnormal cervical cytology in the PAP smear samples has been increased nearly 60% in the UAE when compared with previous studies carried out by Ortashi O and Abdalla D in 2019 (5.3% ASCUS; 0.3% ASC-H) [19], by Fakhreldin M and Elmasry K (3% ASCUS, 0.02% ASC-H out of 9641 samples) [20] and Al-Zaabi *et al.* (2.48% ASCUS, 1.56% LSIL, 0.28% HSIL, 0.1% ASC-H and 92.1% normal cytology) in 2015 [21], and Aswad SG *et al.* 2006 [12]. But in Oman study population, the rate of identified abnormal cytology in PAP smear samples was low (5.3% ASCUS, 1.2% LSIL and 89.5% normal cytology) in the year 2020 [15] and in 2021 (1.8% ASCUS and 1.4% LSIL) [22]. The current prevalence of abnormal cervical cytology was 21.15% and it is being higher than the reports from the neighboring country Oman (6.92% and 3.5%) [15, 22], even from Mexico in 2019 (14.4%) [23] and lower than the reports from Iran (25%) [24]. The current rate was almost nearer to the reports which were from USA (19.6%) and India (19.6%) in 2022 [10, 25]. The current rate was lesser than the reports which were published in 2016 by Sign (46.8%) [26] and in 2013 by Elkharashy *et al.* from Egypt (44.44%) [27].

Prevalence of HPV is found as 60.58% (in abnormal & normal cytology; 17.3% & 43.27%), this finding was lesser than a reported study (88%) which was conducted in 2018 in UAE [16]. The prevalence rate has shown a significant positive awareness of the HPV infection and

vaccination among the population. But in the year 2015 and 2017 [20, 28], the HPV prevalence rate in the UAE was 17.9% and 44.38% respectively, these lower rate might be the reason for lack of cutting-edge techniques like flow through reverse hybridization had not been available to detect the different LR and HR genotypes. The prevalence of HPV infection in the abnormal and normal cytology samples was 27.83% and 72.17% respectively. Similarly, higher prevalence of HPV positivity was detected in the abnormal cytology samples of Oman (37.5%) and UAE (81%) study population. [15, 20].

Within the age group of 20–29-years-old study population, HPV16 & 45 (27.8%), 11 (22.2%), 58 & 67 (16.7%) were the frequently identified. In this age group, HPV11 (16.7%) was the predominant in Arab patients and HPV16 (22.2%) was in non-Arab group. In the age group of 30-39-year-old patients, the most frequently identified genotypes were HPV6 (36%), 45 (16%) and 16 (12%) and in the age group of 40–49, HPV6 (26.3%), 62/81 (21.1%), 11 (15.8%) were the common genotypes. However, the HR-HPV16 & 18 were detected only in the Arab cohort. None of the HPVs were detected in the 50–59-year-old study group except HPV 11 and it was detected only in the non-Arab cohort. Across all the age group of Arab cohort, the HPV6, 11, 45, 16 & HPV67 were the most commonly detectable genotypes and in the non-Arab cohort, the HPV6, 62/81, 16 and 45 were the commonest genotypes. However, the HR-HPV18 was identified in both cohort but its positivity rate was 1.8% and likewise, low positivity rate of HPV18 was detected in the Western Iranian study group [24].

In the present study, the LR-HPV 6 was the most predominant genotype among all the identified HPVs and then HPV16, 45, 11, 62/81 & 67 were the next detectable genotypes. In such a way, the HPV6 was the predominant type, and the others 11,12 & 16 were the next genotype in the Iran study group [24]. But in other previous studies which were conducted in UAE [13], China [29], India [10], Dutch-Caribbean Island [30], Saudi Arabia [31] and Western Mexico [33], the HPV 16 was the predominant one than the other identified genotypes. But in the Oman study population, LR-HPV54 was the predominant genotype followed by HPV82, 42, 68 & 44 [15].

In our study, the higher frequency of HPV infection rate was encountered in the age group of 20–49 years old of both cohort. Similarly, a study conducted in Abu Dhabi, UAE in 2018, stated that the maximum infection rate was observed in the age group between 24–54 years old women [13] and in the year 2016 & 2017, the studies which were conducted in the UAE, mentioned that the maximum HPV infection rate was observed in women over 30-year-old [20, 28]. In Hefei province of China, the maximum infection rate was detected in the age group of 31-50-year-old women [29] and in Western Iran the maximum HPV infection rate was determined in the age group of 31-40-years-old women [24] and in Saudi Arabia, rate was identified in between the age group 30-50-year-old women [31]. The highest HPV positivity rate (85%) was detected in women aged 36–55-year-old from Western Mexico [32]. But some previous studies stated that there was no association between HPV infection and the stratified age groups (≤25, 25–34. 35–44 and ≥45 years) [33]. However, in the developed counties the prevalence of HPV was peak in young women and lowered after 35 years of age [5] and other studies mentioned that there is a second peak in the postmenopausal age groups in some countries [34, 35].

The present study showed that 60.58% of women with cervical infection and/or inflammation had HPV positivity. Among them the high-risk HPVs rate was 14.8%, low-risk genotypes rate was 23.08% and mixed LR and HR genotypes infection rate was 20.3%. Whereas, in Brazil 88.4% of high-risk genotypes was reported and it was significantly higher [36]. The study provided age-specific HPV infection rate among women of Arab and non-Arab cohort. The women aged 20–49 years had the highest infection rate (60.29%). The study found that the prevalence of HPV infection is higher in the younger age group and declined gradually in the

older age group. But in northern Henan Province of China, the rate of HPV infection was higher in 60-year-old women and lower in the middle age group [37].

## Conclusion

The current findings confirm UAE to have a slightly high HPV prevalence. According to the study, LR-HPV6, 11, 67, 62/81 and HR-HPV16, 45 were the most common HPV infections in the women between the age group of 21 and 59-years-old and much less prevalence of HR-HPV18 was found. A moderate increase than expected incidence of HR-HPV45 and 62/81 were detected. Co-infection with multiple low and high-risk genotypes is present in 20.2% of cases. Among them, HPV6 was the most common followed by HPV62/81 and HPV16. Based on the molecular genotyping, 43.27% of the normal epithelia were positive to HPV infections. It is evident that symptomatic women even having normal epithelia has been infected with different low and high-risk genotypes. So, the present study highlights the importance of molecular genotyping to emphasize cervical screening triage. A large population-based study across the UAE is needed to determine the most prominent genotypes and develop new vaccines to reduce the burden of cervical infection.

## Limitations of the study

As of all other studies this study also reported few limitations as follow.

a. The study samples received from various hospitals, clinics and Thumbay hospitals located in and around the northern emirates of UAE (Sharjah, Ajman, Umm-Al-Quwain, and Fujairah) and not received the samples from other emirates (Dubai, Abu Dhabi & Ras-Al-Khaimah).

b. In this HPV Direct Flow CHIP method, the mentioned genotypes HPV62/81, 31/68 & 44/55 could not be able to consider as a single or mixed genotype.

## Supporting information

**S1 File. For each variable of interest, give sources of data and details of methods of assessment (measurement).** Describe comparability of assessment methods if there is more than one group.
(PDF)

**S2 File.** (a) Report numbers of individuals at each stage of study—e.g., numbers potentially eligible, examined for eligibility, confirmed eligible, included in the study, completing follow-up, and analysed.
(PDF)

**S3 File. Flow diagram of the study.**
(PDF)

**S4 File. Cross-sectional study—Report numbers of outcome events or summary measures.**
(PDF)

**S5 File. Cautious overall interpretation of results considering objectives, limitations, multiplicity of analyses, results from similar studies, and other relevant evidence.**
(PDF)

## Acknowledgments

We would like to acknowledge the efforts of Gynaecologists, Pathologists, Genetics and molecular biologists and other relevant staffs of Thumbay Laboratory and Thumbay University Hospital, Ajman, U.A.E for collecting data and conducting this study. Also want to express our gratitude to the College of Medicine, GMU for the constant support to conduct this research.

## Author Contributions

**Conceptualization:** Nazeerullah Rahamathullah.

**Data curation:** Heba Issa Odeh, Sara Rashid Al-badi, Takrim Abdulwali Saeed, Nazeerullah Rahamathullah, Eman Hassan Ibrahim.

**Formal analysis:** Heba Issa Odeh, Sara Rashid Al-badi, May Khalil Ismail, Zahra Arshad Khan.

**Investigation:** Heba Issa Odeh, Sara Rashid Al-badi, Basma Karima, Takrim Abdulwali Saeed.

**Methodology:** Nazeerullah Rahamathullah, Eman Hassan Ibrahim, May Khalil Ismail.

**Project administration:** Nazeerullah Rahamathullah, Eman Hassan Ibrahim, May Khalil Ismail.

**Software:** Heba Issa Odeh, Sara Rashid Al-badi, Basma Karima, Nazeerullah Rahamathullah.

**Supervision:** Nazeerullah Rahamathullah, Eman Hassan Ibrahim, May Khalil Ismail.

**Validation:** Nazeerullah Rahamathullah, Eman Hassan Ibrahim, May Khalil Ismail.

**Writing – original draft:** Heba Issa Odeh, Sara Rashid Al-badi, Basma Karima, Takrim Abdulwali Saeed, Nazeerullah Rahamathullah, Zahra Arshad Khan.

**Writing – review & editing:** Nazeerullah Rahamathullah, Eman Hassan Ibrahim, May Khalil Ismail.

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
