## [Decision Letter · Decision Letter 0]

20 Jun 2023

PONE-D-23-11894Exploring the prevalence of Human Papillomavirus (HPV) genotypes in PAP smear samples of women in northern region of United Arab Emirates (UAE): HPV Direct Flow CHIP system-based pilot studyPLOS ONE

Dear Dr. Rahamathullah,

Thank you for submitting your manuscript to PLOS ONE. After careful consideration, we feel that it has merit but does not fully meet PLOS ONE’s publication criteria as it currently stands. Therefore, we invite you to submit a revised version of the manuscript that addresses the points raised during the review process.

We look forward to receiving your revised manuscript.

Kind regards,

Olatunji Matthew Kolawole, Ph.D.

Academic Editor

PLOS ONE

Additional Editor Comments:

Your manuscript has been well reviewed and we believe it is relevant for publication, however, there are some minor, but crucial, corrections that must be done.

Kindly attend to the comments made by reviewers and present your responses in details.

Reviewers' comments:

Reviewer's Responses to Questions

**Comments to the Author**

1. Is the manuscript technically sound, and do the data support the conclusions?

Reviewer #1: Partly

Reviewer #2: Yes

Reviewer #3: Yes

2. Has the statistical analysis been performed appropriately and rigorously? 

Reviewer #1: Yes

Reviewer #2: Yes

Reviewer #3: Yes

3. Have the authors made all data underlying the findings in their manuscript fully available?

Reviewer #1: Yes

Reviewer #2: Yes

Reviewer #3: Yes

4. Is the manuscript presented in an intelligible fashion and written in standard English?

Reviewer #1: No

Reviewer #2: Yes

Reviewer #3: Yes

5. Review Comments to the Author

Reviewer #1: Abstract

Conclusion should summarize the key learning from the study and implications for practice, policy and research. The results provided in lines 40 to 45 should be moved to Results section.

2.2 Subjects

Line 107 - either describe the laboratory protocol or provide a reference if the protocols are a standard protocol.

2.7 Statistical Analysis

Line 147, it is sufficient to say "data were analyzed", please delete "and statistically evaluated".

3. Results

Please remove leading zeros before integers on lines 156, 173, and 175.

Line 179, start the sentence using "Fifty cases" and not "50 cases".

Table 2 (page 10), Pearson's correlation analysis not described in the Methods section. Please describe all analysis in the Methods section.

References

Lines 390 to 391 (reference 3), please use proper referencing style for a webpage showing date accessed. Recommend using a Reference manager if the references were done manually for consistency.

A general comment, the manuscript will benefit from scientific editing by a native English speaker. For example, the sentence on lines 280 to 283 is unclear. Also, the phrase "Like this" was used too frequently (lines 284, 291, 298 and 301). The language should be varied.

Reviewer #2: I am recommending the artice on Exploring the prevalence of Human Papillomavirus (HPV) genotypes in PAP smear samples of women in northern region of United Arab Emirates (UAE): HPV Direct Flow CHIP system-based pilot study for publication. The research is novel

Reviewer #3: Although the manuscript is technically sound, the analysis well done and presented in an intelligent fashion, the writing standard should be improved on, to make statements in the result and discussion flow for easy understand.

For that, the following points need to be taken into consideration:

INTRODUCTIO:

there should be additional scholarly evidence/reasons for the conduct the research.

METHOD

I suggest:

1. There should be sections explaining;

a. Data Collection

b. study design

2. Study Populations should be used instead of "Subjects".

3. There should be a clear explanation of what was done with the sturdy samples to explain the variation in some reported parameters e.g. the cytology examination result repeated, why is the difference between the first and the second reported cytology result in section 3.1 and 3.2.

4. look at lines 164, 165 and 166 and rephrase the sentence for better understanding.

RESULT

I suggest

1. You should re-write the different sections of the result to make the findings simple to understand.

2. Look again at the tables to make them simpler, and

3. Review the need to use superscript a,b,c,d,e, etc for "n".

DISCUSSION

Discuss the result only, no need to repeat what was already mentioned in the result section. e.g. what is in lines 262 and 263 is a repeat of a statement that is already in the result section

CONCULUSION

Review the conclusion to distinguish between finding from the study and recommendation based on the findings. e.g. the statement in line 340, 341, and 342 contains what looks like a finding from the study but ended up with a recommendation.

6. PLOS authors have the option to publish the peer review history of their article (what does this mean?). If published, this will include your full peer review and any attached files.

Reviewer #1: **Yes: **Oluwasanmi Adedokun

Reviewer #2: **Yes: **Emmanuel Adamlekun

Reviewer #3: **Yes: **Lawan Adamu

---

## [Author Response · Author response to Decision Letter 0]

19 Aug 2023

The following comments mentioned by the reviewers (reviewer 1, 2 &3) are noticed and corrections are done accordingly in the revised manuscript. 

Reviewer # 1 

Abstract

Comment 1: Conclusion should summarize the key learning from the study and implications for practice, policy, and research. The results provided in lines 40 to 45 should be moved to Results section.

Ans: The above-mentioned comments were responded in the conclusion and the changes are highlighted and the results provided in lines 40 to 45 are moved to the results section. 

2.2 Subjects

Comment 2: Line 107 - either describe the laboratory protocol or provide a reference if the protocols are a standard protocol.

Ans: As per the comment, the reference is provided in that section (line 119) and highlighted. 

2.7 Statistical Analysis

Comment 3: Line 147, it is sufficient to say "data were analyzed", please delete "and statistically evaluated"

Ans: The suggestion is added in the line – 159 to 161. 

3. Results

Comment 4: Please remove leading zeros before integers on lines 156, 173, and 175.

Ans: The leading zeros are removed before integers on the lines 167, 189, 191 and highlighted in the revised manuscript.

Comment 5: Line 179, start the sentence using "Fifty cases" and not "50 cases".

Ans: The above-mentioned issue instead of “50 cases”, the word “Fifty cases” is added in the line 195.

Comment 6: Table 2 (page 10), Pearson's correlation analysis not described in the Methods section. Please describe all analysis in the Methods section.

Ans: By mistake, the Pearson’s correlation analysis was mistakenly mentioned in the table 2. Originally, the Pearson’s Chi-square test was performed in tables 2, 3 & 6 with p-value ˂ 0.05 considered as statistically significant. It is mentioned in the revised manuscript on the line 159 to 161 and in the table 2, 3 & 6. 

References

Comment 7: Lines 390 to 391 (reference 3), please use proper referencing style for a webpage showing date accessed. Recommend using a Reference manager if the references were done manually for consistency.

Ans: The proper referencing style is used (3. World Health Organization. Cervical cancer. 2022. https://www.who.int/news-room/fact-sheets/detail/cervical-cancer.) updated in the attached revised copy and highlighted in the line – 404 to 405.

Comment 8: A general comment, 

The manuscript will benefit from scientific editing by a native English speaker. For example, the sentence on lines 280 to 283 is unclear. Also, the phrase "Like this" was used too frequently (lines 284, 291, 298 and 301). The language should be varied.

Ans: The above-mentioned general comments are noticed and the corrections are done accordingly in the discussion part (from the line 279 to 319).

Reviewer# 2

I am recommending the article on Exploring the prevalence of Human Papillomavirus (HPV) genotypes in PAP smear samples of women in northern region of United Arab Emirates (UAE): HPV Direct Flow CHIP system-based pilot study for publication. The research is novel.

Reviewer #3: 

Comment 9: Although the manuscript is technically sound, the analysis well done and presented in an intelligent fashion, the writing standard should be improved on, to make statements in the result and discussion flow for easy understand.

For that, the following points need to be taken into consideration:

INTRODUCTION: there should be additional scholarly evidence/reasons for the conduct the research.

Ans: The above comment is considered and the reasons for the conduct the research is added in the end of the introduction (from the line 86 to 92).

METHOD: 

I suggest:

Comment 10: 1. There should be sections explaining;

a. Data Collection

Ans: The above suggestion is included in the method section of the revised manuscript (line 108 to111) and highlighted.

b. study design

Ans: The above suggestion is incorporated and highlighted in the revised manuscript (from the line 103 to 107).

Comment 11: 2. Study Populations should be used instead of "Subjects".

Ans: The suggestion is added in the revised manuscript (line 112). 

Comment 12: 3. There should be a clear explanation of what was done with the sturdy samples to explain the variation in some reported parameters e.g. the cytology examination result repeated, why is the difference between the first and the second reported cytology result in section 3.1 and 3.2.

Ans: The above-mentioned comment is noticed, and the corrections are done in the result section of the revised manuscript.

3.1. Study samples and Cytological assessment – This title has been updated in to “Cytological Examination”. In this heading the different morphological changes of the cervical PAP smear samples’ results are discussed. How many ASCUS, LSIL, HSIL and ASC-H are identified. How many PAP smear samples have shown normal epithelia even the woman has cervical symptoms, and their details are given in the table 1 and figure 1 & 2. 

3.2. HPV genotype detection – This title is updated into “HPV detection and genotyping” – In this heading, results of how many ASCUS, LSIL, HSIL, ASC-H and normal epithelia are positive to HPV infection, type of the genotypes such as low and high-risk and the predominant HPV genotypes in the samples, and their details are given in the table 2 & 3 and figure 3. The positivity of the HPV infection was done by multiplex Polymerase chain reaction and their genotype was done by flow-through reverse hybridization using HPV direct flow chip kit and it is mentioned in the method section – Flow through reverse hybridization (line 144 – 157). 

Comment 13: Look at lines 164, 165 and 166 and rephrase the sentence for better understanding.

Ans: The above comment is noticed and rephrased the lines (164, 165 and 166) 173 to 174 & 176 to 182 and highlighted in the revised copy. These sentences are rephrased for better understanding. 

RESULT

I suggest

Comment 14: 1. You should re-write the different sections of the result to make the findings simple to understand. 

Ans: Yes, the result section has been written and rephrased to make the findings simple to understand. 

Comment 15: 2. Look again at the tables to make them simpler,

Ans: The tables 1, 2, 3, 4, 5 & 6 are simplified and highlighted in the simplified areas. 

Comment 16: 3. Review the need to use superscript a,b,c,d,e, etc for "n".

Ans: All the superscripts mentioned in the tables (1,2,3,4,5 & 6) are removed, simplified, and highlighted in the revised manuscript. 

DISCUSSION

Comment 17: Discuss the result only, no need to repeat what was already mentioned in the result section. e.g. what is in lines 262 and 263 is a repeat of a statement that is already in the result section. 

Ans: The above-mentioned comment is noticed, and the corrections are done accordingly at the lines 278 -286, 293 -294, 295 -300, 302, 306, 313, 317 -318, 334 in the revised manuscript. 

CONCLUSION

Comment 18: Review the conclusion to distinguish between finding from the study and recommendation based on the findings. e.g. the statement in line 340, 341, and 342 contains what looks like a finding from the study but ended up with a recommendation.

Ans: As per the comment, the conclusion is revised in the updated manuscript and highlighted (Lines 355 to 360).

Reference section

The references 27, 33 & 37 are arranged as per the journal’s reference style and highlighted in the revised manuscript.

---

## [Editor Report · Decision Letter 1]

21 Aug 2023

Exploring the prevalence of Human Papillomavirus (HPV) genotypes in PAP smear samples of women in northern region of United Arab Emirates (UAE): HPV Direct Flow CHIP system-based pilot study

PONE-D-23-11894R1

Dear Dr. Rahamathullah,

We’re pleased to inform you that your manuscript has been judged scientifically suitable for publication and will be formally accepted for publication once it meets all outstanding technical requirements.

Kind regards,

Olatunji Matthew Kolawole, Ph.D.

Academic Editor

PLOS ONE
---

## [Editor Report · Acceptance letter]

25 Aug 2023

PONE-D-23-11894R1 

*Exploring the prevalence of Human Papillomavirus (HPV) genotypes in PAP smear samples of women in northern region of United Arab Emirates (UAE): HPV Direct Flow CHIP system-based pilot study*

Dear Dr. Rahamathullah:

I'm pleased to inform you that your manuscript has been deemed suitable for publication in PLOS ONE. Congratulations! Your manuscript is now with our production department. 

Kind regards, 

on behalf of

Dr. Olatunji Matthew Kolawole 

Academic Editor

PLOS ONE